# HIVE-3D: Hierarchical Voxel Enhancement for High-Quality 3D Scene Generation

**Bin Zang** [* 1]  **Wenting Zheng** [* 1]  **Xiaoliang Luo** [2]  **Zhiyuan Fang** [1]  **Shi Li** [1]  **Lvchun Wang** [2]  **Wei Yu** [2]  **Yi Zhao** [2]  **Tian Xie** [1]  **Yuchi Huo** [1]  **Rengan Xie** [1]

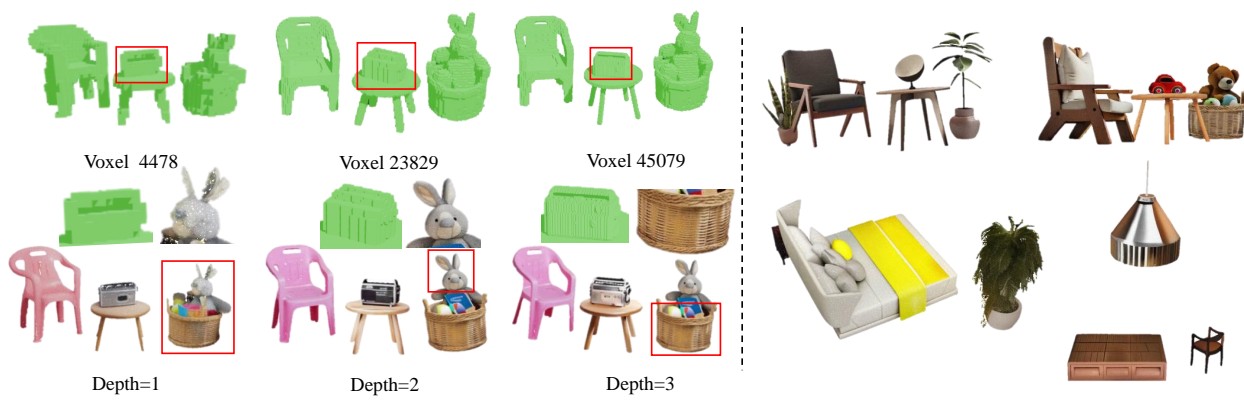

(a) Progressive 3D Resolution Enhancement                    (b) Generated 3D Scenes

*Figure 1.* HIVE-3D generates high-resolution, high-fidelity 3D scenes from a single image using a hierarchical voxel enhancement framework within a coarse-to-fine scheme.

## Abstract

Recently, a line of works can generate impressive 3D objects from a single image, but they are limited by restricted representation resolution, making them unsuitable for 3D scene generation. In this work, we introduce HIVE-3D, a novel method for high-quality 3D scene generation based on hierarchical voxel enhancement framework. Specifically, given a single scene image as input, we first produce a coarse initial scene, then introduce image segmentation and attention-based retrieval to align 2D image components with 3D scene components. Subsequently, we organize these scene relations into a hierarchical component tree, where nodes closer to the leaves denote finer-grained components. Finally, we propose a voxel super-resolution model that generates refined voxels for the target instance while maintaining strong consistency with the coarse voxels. Equipped with this model, we perform coarse-to-fine hierarchical super-resolution on images and voxels for each component, producing a high-resolution and high-quality 3D scene. Extensive experiments demonstrate that our method significantly outperforms previous approaches, achieving state-of-the-art performance.

## 1 Introduction

Rapid generation of a 3D scene from a single image is strongly desired across games, film, and industry. However, current scene generation methods (Gao et al., 2024; Chen et al., 2024a; Nie et al., 2020; Ardelean et al., 2025; Meng et al., 2025) still struggle with complex layouts and diverse, personalized objects, leaving high-quality 3D scene synthesis an open challenge.

Retrieval-based methods (Gao et al., 2024; Gumeli et al., 2022; Izadinia et al., 2017) are a classic solution for 3D scene generation. Such methods aim to build a comprehensive CAD template library and treat generation as retrieving the best-matching objects for an input RGB image and assembling them into a spatial layout. However, due to limited template diversity, these methods fail to align precisely with

*Equal contribution [1]State Key Laboratory of CAD&CG, Zhejiang University, Hangzhou, China [2]China Mobile (Jiangxi) Virtual Reality Technology Co., Ltd., Nanchang, China. Correspondence to: Rengan Xie <rgxie@zju.edu.cn>.

*Proceedings of the 43rd International Conference on Machine Learning*, Seoul, South Korea. PMLR 306, 2026. Copyright 2026 by the author(s).

object shapes in the input image and cannot produce textured scenes. To obtain more faithful 3D scene reconstructions, subsequent research (Nie et al., 2020; Paschalidou et al., 2021; Zhang et al., 2021) leverages neural networks to learn end-to-end mappings from a single input image to complete 3D scenes. These models jointly infer geometry, layout, and texture, enabling holistic scene generation with improved accuracy and robustness. However, generating the entire scene in a single inference inherently limits spatial resolution and detail fidelity. In addition, the scarcity of large-scale, high-quality 3D scene data for training limits the generalization.

To address this issue, an effective approach is to decompose the 3D scene into multiple single-object generations and then assemble them into a coherent layout, which has given rise to a series of compositional methods (Chen et al., 2024b; Ardelean et al., 2025; Huang et al., 2025; Dong et al., 2025). These works can produce more detailed 3D scenes, and thanks to the generalization of single-object 3D generative models, the range of applicable 3D scenes has been further expanded. However, these methods often perform a single pass of generation and assembly for each component in a scene, without considering that different components require varying levels of detail and refinement. Building on these observations, we propose a novel hierarchical voxel enhancement framework for 3D scene generation, which progressively enhances the geometric resolution of each local component in a coarse-to-fine scheme and produces high-fidelity 3D scenes.

First, given an input scene image, we first employ TRELLIS (Xiang et al., 2025) to produce an initial scene. Constrained by the resolution of the TRELLIS model, this initial result is relatively coarse, yet it retains globally coherent geometry and layout information. Subsequently, we aim to dissect the target scene hierarchically and construct a hierarchical scene tree. However, performing instance-level segmentation directly on the 3D initial scene remains nontrivial. To address this issue, we leverage an image-based vision model to perform hierarchical partitioning on the image, and we introduce a 2D-to-3D matching strategy to lift the hierarchical segmentation into the 3D scene space, yielding component images and their corresponding local scene components. Furthermore, we aim to refine coarse scene components while preserving feature consistency. A naive strategy applies an image super resolution model and feeds the enhanced image into the TRELLIS model for higher resolution representations, but the stochasticity of generative models induces deviations from the coarse input, undermining hierarchical generation. To this end, we propose the voxel super resolution model, which takes the coarse voxel and the fine image as conditions to generate a refined voxel with higher resolution and richer geometric detail. Finally, we perform a top-down hierarchical enhancement of geo-

metric fidelity for components in the scene tree, introduce a shape-based strategy to align each regenerated component with its coarse counterpart, and register it back into the original scene, yielding the final high-resolution 3D scene.

Overall, the main contributions of this work can be summarized as follows.

- We propose HIVE-3D, a novel hierarchical pipeline that progressively generates high-resolution 3D scenes from a single RGB image.

- We introduce a 2D-to-3D matching strategy that lifts hierarchical 2D image segmentation into 3D scene space, producing a hierarchical 3D scene tree.

- We propose a voxel super-resolution model that enables increasing the resolution of target 3D components while maintaining consistent features with the coarse voxels clues.

## 2 Related Work

### 2.1 Image-to-3D Scene Generation

**Retrieval-based methods.** A distinct paradigm in 3D scene generation leverages retrieval from extensive databases of CAD models (Gao et al., 2024; Gumeli et al., 2022; Izadinia et al., 2017; Kuo et al., 2020; 2021; Langer et al., 2022). Instead of synthesizing geometry from scratch, this approach composes scenes by identifying and assembling pre-existing assets. While this guarantees high-fidelity components, it introduces fundamental challenges in expressiveness and composition. The expressiveness is bounded by the diversity of the underlying library, hindering generalization to novel scenes. On the other hand, the compositional challenge stems from the mismatch between CAD templates and real objects, making a globally coherent spatial arrangement a difficult combinatorial problem.

**Feed-forward scene reconstruction.** A significant category of 3D scene reconstruction methods relies on end-to-end training with 3D supervision (Chen et al., 2024a; Chu et al., 2023; Dahnert et al., 2021; Gkioxari et al., 2022; Liu et al., 2022; Nie et al., 2020; Paschalidou et al., 2021; Zhang et al., 2021; 2023b; Xiang et al., 2025; Yao et al., 2025). These approaches typically employ an encoder-decoder architecture to directly regress a complete 3D scene representation, such as a voxel grid with geometry and instance labels, from a single input image. The primary advantage of this holistic approach is that by jointly predicting the scene layout and its contained objects, the relative object poses are intrinsically correct. However, these methods exhibit low output resolution at the component level and suffer poor generalization to out-of-distribution scenes, as large-scale 3D scene datasets are hard to obtain for training.

**Compositional generation methods.** A prominent direction in 3D scene generation is the compositional approach (Chen et al., 2024b; Ardelean et al., 2025; Han et al., 2025; Zhou et al., 2024), which leverages powerful pre-trained models to reconstruct scenes in a modular, multi-stage pipeline. These methods typically first decompose a scene from an input image (Ardelean et al., 2025; Dong et al., 2025; Meng et al., 2025), then employ specialized models for tasks like object completion, per-object 3D generation, and finally optimize the spatial layout to reassemble the scene. This modularity enhances generalization by capitalizing on large-scale generative priors. In contrast to holistic generation approaches (Xiang et al., 2025; Yao et al., 2025; Zhang et al., 2021), compositional generation methods excel at producing high-quality object instances. However, they often struggle to preserve global scene context, making spatial alignment a significant challenge. Our approach bridges this gap by first generating the scene holistically to establish a coherent layout, and then hierarchically refining each instance while preserving this overall structure. This results in a scene that is both high-resolution and globally consistent.

## 2.2 Model Adaptation Methods

With the rise of large-scale, pre-trained generative models (Xiang et al., 2025; Yao et al., 2025; Huang et al., 2025; Li et al., 2024; Wu et al., 2024; Zhang et al., 2024; Zhao et al., 2023), parameter-efficient fine-tuning (PEFT) has emerged as a crucial paradigm for adapting these foundational models to new tasks without incurring the prohibitive costs of full fine-tuning. A prominent strategy in this domain is to introduce small, trainable modules while keeping the original model weights frozen. Prominent PEFT strategies adapt frozen models by introducing small, trainable modules, such as ControlNet (Zhang et al., 2023a), which adds a trainable copy of network blocks to process spatial conditions, and LoRA (Hu et al., 2022), which injects low-rank matrices into existing layers based on the hypothesis that weight updates have a low intrinsic rank.

Building on similar principles, IP-Adapter (Ye et al., 2023) focuses on enabling image-prompting capabilities. It introduces a lightweight adapter module that uses a cross-attention mechanism to inject visual features, extracted by a separate image encoder, into a frozen text-to-image diffusion model (Ho et al., 2020), achieving high-fidelity image-conditioned generation in a decoupled manner. Inspired by IP-Adapter, we implement our voxel super-resolution model by introducing a similar fine-tuning strategy to the flow transformer of TRELLIS. Specifically, we treat the coarse voxels as an additional condition to guide the generation of refined voxels, thereby achieving high-resolution output while maintaining structural consistency.

## 3 Preliminary

Our approach builds on a state-of-the-art 3D generative model TRELLIS (Xiang et al., 2025), where the target is represented as structured latent voxels. For a 3D asset $\mathcal{O}$, TRELLIS encode its geometry and appearance information using a unified structured latent representation $\boldsymbol{z}$, as:

$$\boldsymbol{z} = \{(\boldsymbol{z}_i, \boldsymbol{p}_i)\}_{i=1}^L, \boldsymbol{z}_i \in \mathbb{R}^C, \boldsymbol{p}_i \in \{0, 1, \dots, N-1\}^3, \quad (1)$$

where $\boldsymbol{p}_i$ is the positional index of an active voxel in the 3D grid, $\boldsymbol{z}_i$ denotes a local latent attached to the corresponding voxel $\boldsymbol{p}_i$. L is the total number of active voxel grid, and N is the spatial length of the voxel grid. Collectively, these structured latents fully describe the surface of $\boldsymbol{O}$, capturing its global structure and fine-grained details in a unified representation.

TRELLIS generates these structured latents using a two-stage pipeline. In the first stage, the model uses a sparse structure flow transformer model $\mathcal{G}_{\mathrm{S}}$ to generate a low-resolution feature grid $\boldsymbol{S} \in \mathbb{R}^{D \times D \times D \times C_{\mathrm{S}}}$, followed by a latent feature decoder $\mathcal{D}_{\mathrm{S}}$ to obtain a dense binary voxel grid $\boldsymbol{O} \in \{0, 1\}^{N \times N \times N}$. This dense grid $\boldsymbol{O}$ is then processed into a set of sparse voxel coordinates $\{\boldsymbol{p}_i\}_{i=1}^L$. In the second stage, TRELLIS employs a subsequent structure latent flow transformer $\mathcal{G}_{\mathrm{L}}$ to generate a set of structural features, $\{\boldsymbol{z}_i\}_{i=1}^L$, corresponding to each of the sparse voxel coordinates from the previous stage. The structured latent $\boldsymbol{z}$ is then processed through specialized decoders($\mathcal{D}_{\mathrm{NeRF}}, \mathcal{D}_{\mathrm{Mesh}}$, or $\mathcal{D}_{\mathrm{GS}}$) to generate the final 3D asset $\mathcal{O}$ in various formats(NeRF, meshes, or 3DGS).

## 4 Method

In this paper, we propose HIVE-3D, a novel pipeline aims to generates high-resolution 3D scenes from a single RGB image. As shown in Figure 2, Our approach decomposes the hierarchical structure of target scene and constructs a scene tree $\mathcal{T}_d$ with depth $d$, where each node denotes a scene component that includes its corresponding voxel representation and cropped image. We denote these components and images as $\{\mathcal{V}_k^d\}_{k=1}^K$ and $\{\mathcal{I}_k^d\}_{k=1}^K$, indicating that there are components at level $d$. Nodes with larger depth $d$ focus on smaller local regions and exhibit finer geometry and texture. In particular, to obtain hierarchical scene components and their corresponding images, we propose a 2D-to-3D scene tree construction method as shown in Section 4.1. On the other hand, to preserve cross-level component consistency, we propose a voxel super-resolution model in Section 4.2 that refines the target instance while remaining consistent with the coarse voxels. Equipped with this model, we introduce the coarse-to-fine hierarchical scene generation as described in Section 4.3, which aims to progressively refine the target component and assemble it back into the scene, producing a high-resolution and high-quality 3D scene.

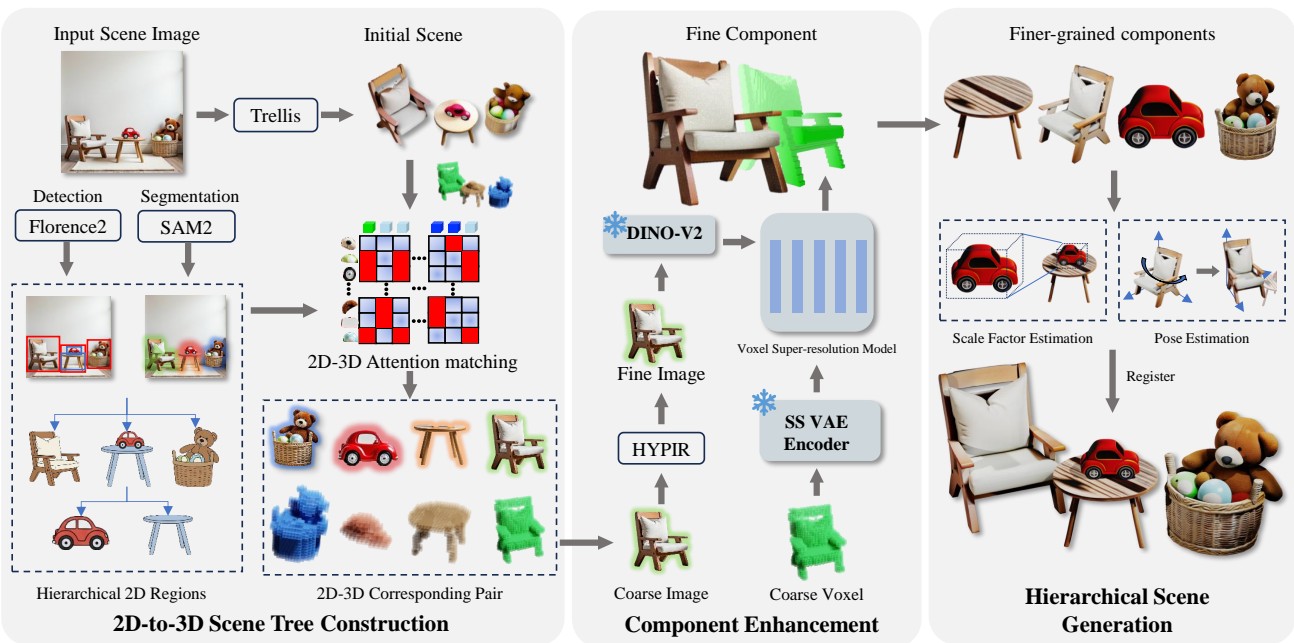

*Figure 2.* **Overview of HIVE-3D.** Our method first produces a coarse 3D scene. We then construct a hierarchical scene tree by decomposing the scene and aligning 2D image parts with 3D voxel components using segmentation and attention-based retrieval. To enhance detail, we propose a voxel super-resolution model that refines each component while maintaining consistency with its coarser representation. Finally, guided by the hierarchical scene tree, we progressively refine the output in a coarse-to-fine fashion, yielding a high-resolution and high-quality 3D scene.

## 4.1 2D-to-3D Scene Tree Construction

We initialize the scene tree $\mathcal{T}_d$ by generating the root node from the input RGB image $\mathcal{I}_{input}$ using TRELLIS (Xiang et al., 2025), which yields initial scenes with coherent spatial structure.

Further, the central focus of our subsequent work is to preserve this plausible spatial arrangement while refining the geometry and texture of each individual component and instance. However, performing instance-level segmentation directly on the voxels of the initial scene is a non-trivial task. Fortunately, inspired by Fuse3D (Jin et al., 2025), which observed that the flow transformer-based model $\mathcal{G}_L$ within the TRELLIS framework learns to establish correspondences between image regions and 3D voxels during its training process. This key insight enables us to recast the challenging task of 3D voxel segmentation as a more tractable problem of performing instance segmentation in 2D, where mature and powerful models are readily available.

Specifically, for any scene component image $\mathcal{I}_k^d$, we employ the object detection model Florence-2 (Xiao et al., 2024) to produce axis-aligned bounding boxes for each detected instance. Subsequently, these bounding boxes are used as prompts for the Segment Anything 2 (SAM 2) (Ravi et al., 2025) model to perform instance segmentation on the $\mathcal{I}_k^d$. Benefiting from the support of SAM2 for hierarchical segmentation, this process yields a precise mask $\{\mathcal{M}_k^d\}_{k=1}^K$

and the corresponding cropped image $\{\mathcal{I}_k^d\}_{k=1}^K$ for each component across levels of scene tree $\mathcal{T}_d$, where K denotes the total number of detected instances at level $d$.

On the other hand, in generating initial active voxels for a component scene in $\mathcal{G}_L$ condition on image $\mathcal{I}_k^d$, image tokens are injected into cross-attention layers as keys and values, while the active voxels serve as queries. This produces a cross-attention map that is normalized along the image token axis using a softmax function. Consequently, the attention score attached to each voxel represents its degree of alignment with the corresponding image tokens, and a larger score indicates stronger correspondence between them. Next, we aggregate the attention scores of image tokens covered by the instance mask region $\mathcal{M}_k^d$. By applying a threshold to these aggregated scores, we select a set of voxel indices denoted as $\mathcal{V}_k^d$, which correspond to the 3D instance aligned with the image region $\mathcal{I}_k^d$. This process is repeated independently for each instance. By applying these indices to the global structure latents, we obtain a set of instance-specific structure latents $\{(\boldsymbol{z}_i, \boldsymbol{p}_i) | i \in \mathcal{V}_k^d\}_{k=1}^K$. The structure latents corresponding to each instance can be further independently decoded into a 3D Gaussian Splatting (3DGS) representation of the object instance, denoted as $\mathcal{O}_k$, using the TRELLIS structure latents decoder $\mathcal{D}_{GS}$.

Finally, we construct a hierarchical scene tree for the input image, with nodes corresponding to scene components at

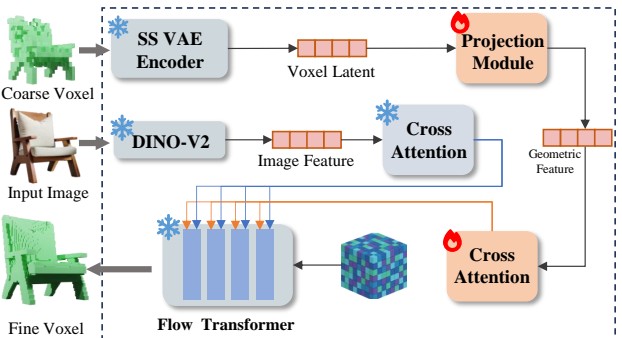

*Figure 3.* **The network structures for voxel super-resolution model.**

different granularities along with the image description. Yet the current tree contains only initial scene voxels, which are relatively coarse. Enhancing their resolution while preserving global consistency remains challenging.

## 4.2 Voxel Super-resolution Model

In this section, we introduce a method to enhance the resolution and quality of coarse scene component while preserving consistent features. A straightforward approach is to apply an image super resolution model HYPIR (Lin et al., 2025) to the image corresponding to the coarse component, then feed the enhanced image back into the TRELLIS model to generate higher resolution representations. However, due to the inherent stochasticity of generative models, the regenerated components often deviate from the coarse input, and this inconsistency impedes hierarchical generation.

To address this issue, we propose the voxel super-resolution model, which introduces the coarse voxel as a new condition within the original generative model $\mathcal{G}_S$ from TRELLIS to generate a refined voxel with higher resolution and richer geometric detail. However, the pretrained model $\mathcal{G}_S$ is optimized for image conditioning, which makes it nontrivial to enable it to accept an additional coarse voxel conditioning with fundamentally different features. Inspired by IP-Adapter (Ye et al., 2023), we aim to train an adapter network that enables it to accept encoded coarse voxels as a condition and inject the projected conditional features into the DiT-based pretrained model $\mathcal{G}_S$ via attention. As shown in Figure 3, we use the pretrained sparse-structure VAE encoder from the TRELLIS model to encode coarse voxels into a latent feature, aligning the voxel condition with the latent space in which the DiT conducts diffusion inference. Further, we add a trainable linear layer as a projection module to map the new geometric condition, aligning it with the original image condition features. Finally, we augment the original DiT block with a dedicated trainable cross-attention layer to inject the new geometric condition feature. Notably, the geometric condition feature and the original image condi-

tion jointly control the DiT block. Details on the attention computation can be found in the supplementary material.

During training, we keep the parameters of the $\mathcal{G}_S$ frozen and exclusively train the newly added cross-attention layers and the projection module. We adopt the same training objective as the $\mathcal{G}_S$:

$$\mathcal{L}_{CFM}(\theta) = \mathbb{E}_{t,\boldsymbol{x}_0,\boldsymbol{\epsilon}}\|\boldsymbol{v}_\theta(\boldsymbol{x},t) - (\boldsymbol{\epsilon} - \boldsymbol{x}_0)\|_2^2, \quad (2)$$

where $\boldsymbol{x}_0$ represents the data samples, $\epsilon$ denotes noise samples drawn from a prior distribution (typically a standard normal distribution), and $t$ is the timestep.

## 4.3 Hierarchical Scene Generation

By modeling the 3D scene with a hierarchical representation and equipping it with a voxel super-resolution module, we can then perform top-down coarse-to-fine scene generation. Specifically, the root node of the scene tree represents the entire initial scene. We then hierarchically partition higher-level nodes into several child components to form child nodes. Using the coarse voxels from the parent node as a condition, we generate finer-grained child components. The rationale and necessity behind the recursive generation strategy are further elaborated in Section C.2. In particular, the segmented child components may be occluded, and we adopt the Amodal3R (Wu et al., 2025) framework to generate complete amodal reconstructions free from occlusion. The above procedure is applied recursively to each node until the desired level of detail is reached.

Nevertheless, due to the structural priors learned from large-scale 3D datasets, the voxel super-resolution model inherently reconstructs these refined components in a canonical reference frame, resulting in a normalized scale and a standardized orientation. Consequently, these high-resolution components exhibit substantial discrepancies in scale and pose relative to the initial scene. To this end, we must align the scale and pose of the finer-grained components with their counterparts in the initial scene, involving the scaling factor $\boldsymbol{s}$ and a rigid transformation with rotation $\mathcal{R}$ and translation $\boldsymbol{t}$.

For the estimation of the scaling factor $\boldsymbol{s}$, we devise a shape-based strategy. Specifically, we uniformly sample points from the surface of the Gaussian splatting representation and compute the average distance of these points to the centroid of the object. This average distance serves as a robust proxy for the overall scale of the object, which we then use to align the scales of different objects. More details are provided in the supplementary material. On the other hand, since all components are represented by 3D Gaussian Splatting 3DGS and each Gaussian primitive can be treated as a point augmented with rich attributes, we leverage point cloud registration to estimate the rigid transformation with

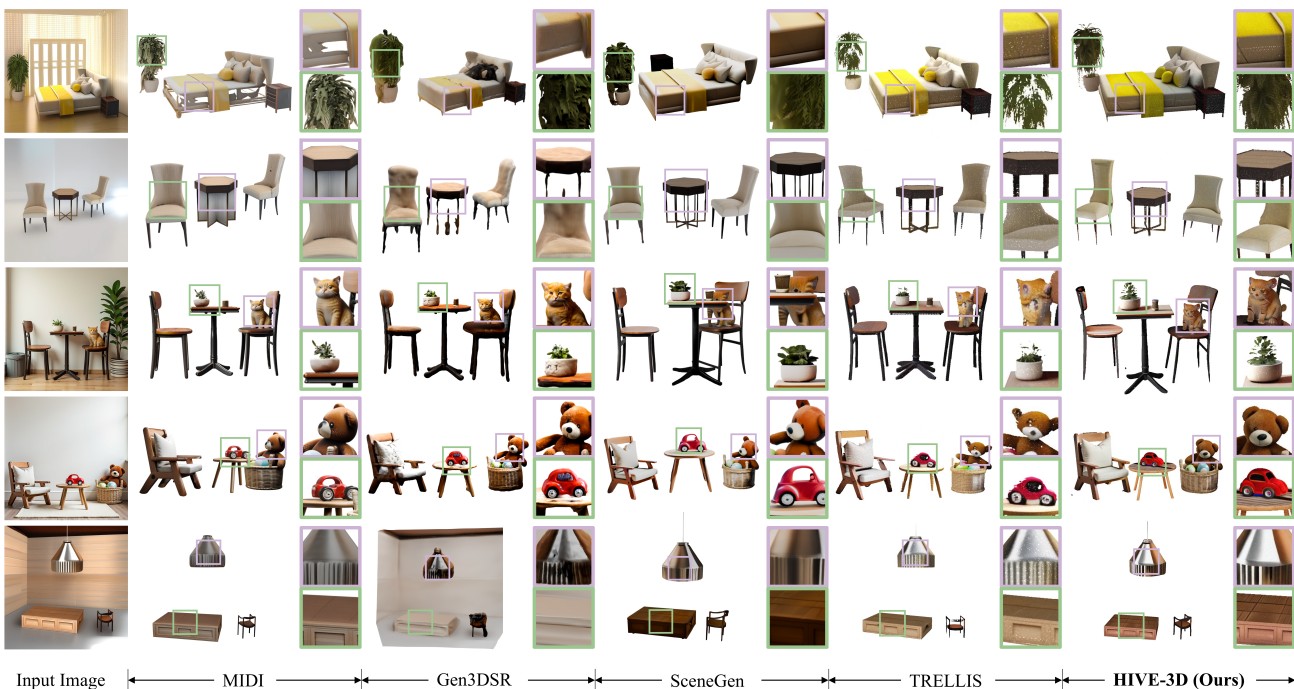

*Figure 4.* **Generation quality comparison with previous methods(Zoom in for details).**

rotation $\mathcal{R}$ and translation $t$. A critical challenge is the severe imbalance in the number of Gaussians between the finer-grained child components and its low-resolution counterpart, which induces substantial mismatches and outliers. To address this, we adopt a registration strategy that emphasizes robustness to outliers rather than strict point-to-point fidelity. Specifically, we employ RANSAC (Fischler & Bolles, 1987), which is resilient to spurious correspondences, instead of methods such as ICP (Besl & Mckay, 1992) that rely on accurate nearest-neighbor matching. The comparison of the two methods is shown in Figure 7. The details of our RANSAC pipeline are provided in the supplementary material.

## 5 Experiment

### 5.1 Setup

**Implementation details.** Our voxel super-resolution model is built upon the flow transformer $\mathcal{G}_S$ of TRELLIS. It is designed to be conditioned on two distinct inputs to control the voxel generation process: an image of 518x518 resolution and a coarse voxel grid. For the image condition, similar to TRELLIS (Xiang et al., 2025), we utilize a DI-NOv2 (Oquab et al., 2023) model to extract visual features. For the coarse voxel condition, we employ the sparse VAE encoder to project it into latent space.

Inspired by IP-Adapter (Ye et al., 2023), we propose a training strategy for our voxel super-resolution model. We freeze

the original parameters of $\mathcal{G}_S$ and integrate a new cross-attention layer into each block. These new layers match the dimensions of the original ones and are initialized with the weights of their corresponding pre-trained counterparts. During training, we exclusively update the parameters of the new projection module and cross-attention layers. The model is trained with 4 NVIDIA 4090 GPUs.

**Datasets.** We trained our voxel super-resolution model on a curated subset of 10,000 3D assets from Objaverse-XL (Deitke et al., 2023). For each asset, we first rendered 24 multi-view images and then downsampled them to generate corresponding coarse voxels. This process yielded a final training set composed of image-voxel pairs.

**Baselines.** We compare our method with the state-of-the-art methods in scene generation from single images, which include feed-forward generation methods TREL-LIS (Xiang et al., 2025), MIDI (Huang et al., 2025), SceneGen (Meng et al., 2025) and compositional generation method Gen3DSR (Ardelean et al., 2025). Furthermore, a detailed comparison with PartPacker (Tang et al., 2026) is provided in Section C.3.

### 5.2 Comparison Results.

**Qualitative Results.** We further evaluated our method using rendered images from the 3D-FRONT (Fu et al., 2021) dataset and a collection of real-world scene photographs.

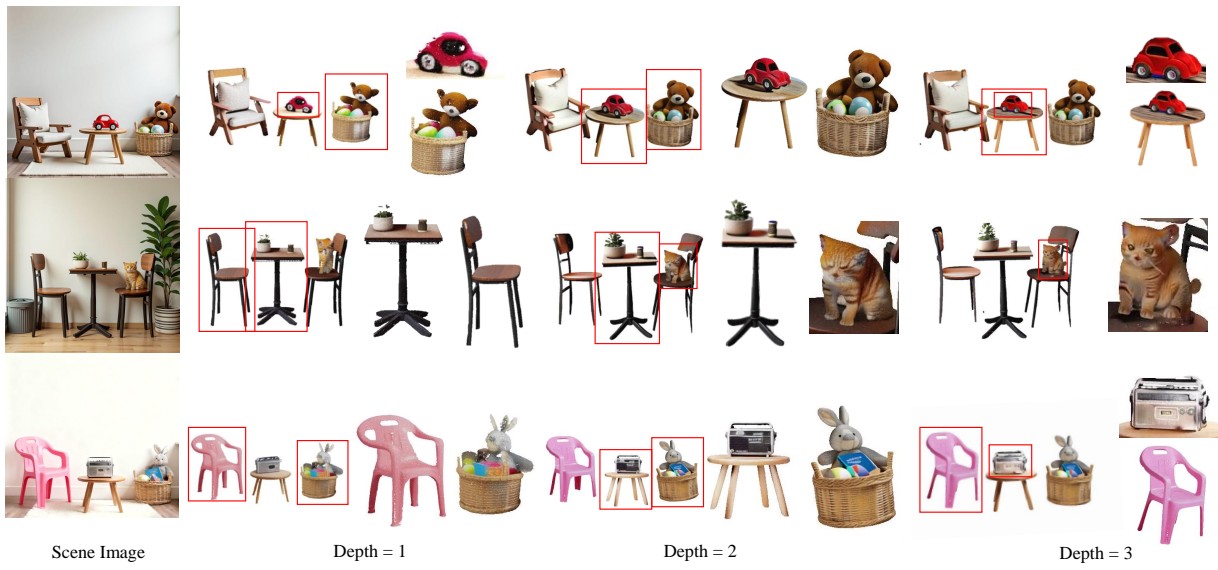

Scene Image   Depth = 1   Depth = 2   Depth = 3

*Figure 5.* **Scene generation results under different recursion depths(Zoom in for details).**

The qualitative results are presented in Figure 4. While existing methods demonstrate certain strengths, they often suffer from significant limitations. TRELLIS excels at generating coherent spatial layouts, but its reliance on a low-resolution voxel representation prevents the synthesis of high-resolution 3D scenes (e.g., Figure 4, third row, fourth column; and fourth row, fourth column). Gen3DSR (Ardelean et al., 2025) performs modestly in rendering geometric and texture details, with its outputs often plagued by object interpenetration and surface distortions as shown in Figure 4 (second row, second column and fourth row, second column). Similarly, scenes from SceneGen (Meng et al., 2025) frequently exhibit severe interpenetration or object displacement artifacts (e.g., Figure 4, third row, third column; and fifth row, third column). Akin to TRELLIS, MIDI (Huang et al., 2025) also produces scenes with plausible layouts but fails to achieve the desired quality in the geometry and texture of individual assets as shown in Figure 4 (first row, first column and second row, first column). In contrast, our method not only inherits the superior layout understanding of TRELLIS but also dramatically enhances the geometric detail and texture fidelity of in-scene instances, enabling the generation of high-resolution, high-quality 3D scenes.

**Quantitative Results.** We evaluate our method and all baselines on the 3D-FRONT dataset using CD (Fan et al., 2017), F-Score (Tatarchenko et al., 2019), and IoU (Choy et al., 2016) to assess the geometric fidelity of the generated scenes. As shown in Table 1, HIVE-3D demonstrates consistently strong performance across all metrics. We also report the average runtime for each method to generate one scene. In addition, we conduct complementary evaluations on both synthetic datasets and real-world im-

*Table 1.* Quantitative Comparisons on Geometry.

| Method | CD↓ | F-Score↑ | IoU↑ | Runtime↓ |
|---|---|---|---|---|
| Gen3DSR | 0.429 | 5.096 | 0.033 | 9min |
| MIDI | 0.056 | 19.83 | 0.5329 | 40s |
| SceneGen | 0.116 | 18.95 | 0.0587 | 29s |
| TRELLIS | 0.0038 | 81.29 | **0.8603** | 6.3s |
| Ours | **0.0035** | **84.34** | 0.7449 | 36.5s |

ages using SSIM, PSNR, LPIPS (Zhang et al., 2018), and CLIP (Radford et al., 2021) to further assess the visual quality and text-image consistency of the generated results. The detailed results are provided in Appendix Table 6.

### 5.3 Ablation Study.

To validate the efficacy of each core component in HIVE-3D, we perform comprehensive ablation studies. Our analysis isolates the effects of four key aspects: 1) the depth of the hierarchical generation, 2) the voxel super-resolution module, 3) the scale factor estimation, and 4) the RANSAC-based point cloud registration (Fischler & Bolles, 1987). We present the quantitative results of our ablation study, evaluated with the SSIM, PSNR, LPIPS (Zhang et al., 2018), CLIP (Radford et al., 2021). More ablation studies are provided in the supplementary material.

**Depth of the hierarchical generation.** We evaluated the effect of varying the number of recursive layers by testing our generation process with depths of 1, 2, and 3. As illustrated in Figure 5, the quality and level of detail in the generated scenes progressively improve as the recursion depth increases. To quantitatively validate this observation,

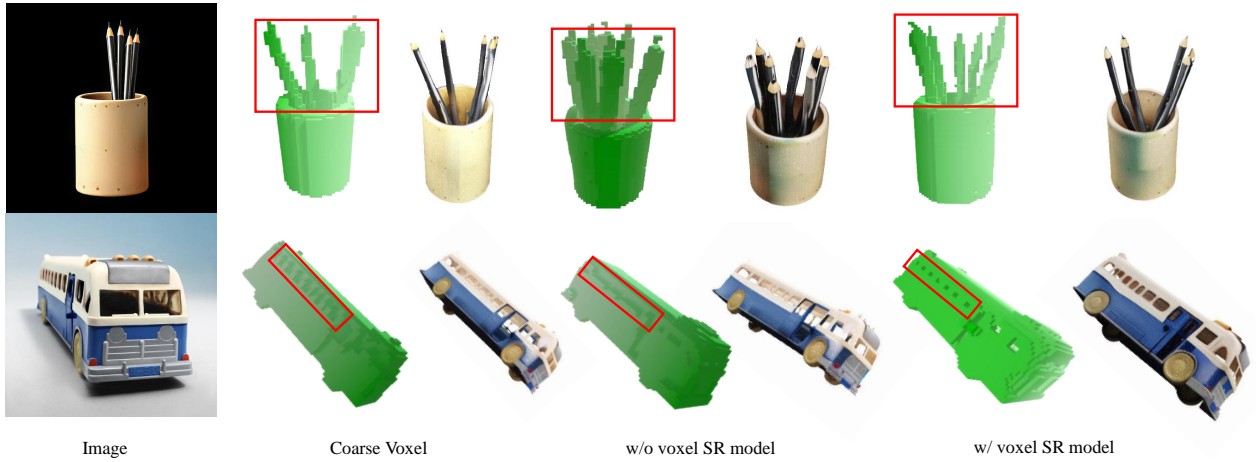

| Image | Coarse Voxel | w/o voxel SR model | w/ voxel SR model |

*Figure 6.* **Impact of coarse voxels on the generation of instance voxels.**

*Table 2.* Ablation study on the depth of the scene tree.

| Method | SSIM↑ | PSNR↑ | LPIPS↓ | CLIP↑ |
|--------|-------|-------|--------|-------|
| Depth=1 | 0.70 | 10.13 | **0.38** | 0.94 |
| Depth=2 | 0.72 | 11.20 | 0.41 | 0.95 |
| Depth=3 | **0.75** | **11.89** | 0.41 | **0.96** |

Table 2 presents a detailed analysis of performance metrics as a function of the generation depth.

**Voxel super-resolution model.** Unlike methods that directly generate voxels conditioned on an image, our approach utilizes coarse voxels to guide the generation process. This allows the resulting super-resolved voxels to retain key structural features from the coarse input, as demonstrated in Figure 6. The quantitative results for our voxel super-resolution model are shown in Table 3. Our model achieves the best scores across all evaluated metrics, including PSNR, SSIM, CLIP and LPIPS. This result demonstrates the superior capability of our approach in reconstructing high-fidelity details from coarse voxel inputs while maintaining structural consistency.

**Scale factor estimation.** Failing to estimate the scale of locally generated scenes or instances from our instance generation module leads to two significant issues. First, it results in noticeable scale inconsistencies across different parts of the scene. Second, and more critically, it severely impairs the subsequent point cloud registration process, as estimating a reliable rigid transformation matrix between two geometries with large scale discrepancies is often infeasible. Figure 7 visually demonstrates the scale artifacts, while Table 4 quantitatively reports the severe drop in registration performance.

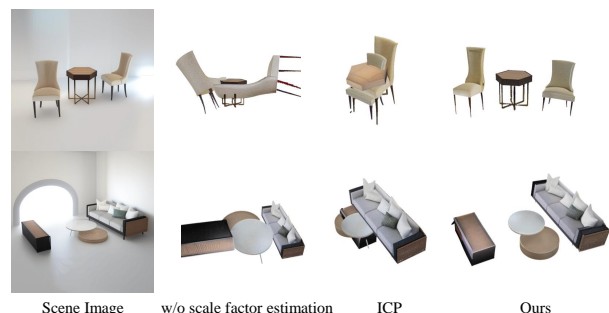

| Scene Image | w/o scale factor estimation | ICP | Ours |

*Figure 7.* **The effect of the scale factor on scene.**

*Table 3.* Ablation study on the voxel SR model.

| Method | SSIM↑ | PSNR↑ | LPIPS↓ | CLIP↑ |
|--------|-------|-------|--------|-------|
| w/o voxel SR model | 0.72 | 10.59 | 0.38 | 0.93 |
| w/ voxel SR model | **0.76** | **11.61** | **0.35** | **0.96** |

**RANSAC-based point cloud registration.** Estimating the transformation between two geometries that exhibit significant differences in point density and slight variations in shape presents a considerable challenge. We found that many precision-oriented registration methods, such as ICP (Besl & Mckay, 1992), failed to produce satisfactory results under these conditions. Ultimately, we adopted the RANSAC algorithm (Fischler & Bolles, 1987), which is known for its strong robustness against outliers and noise, and successfully achieved the desired alignment. We justify this algorithmic choice with both qualitative and quantitative evidence. Figure 7 visually contrasts a successful alignment by RANSAC with a typical failure case from ICP. Concurrently, Table 4 provides a quantitative comparison, demonstrating that RANSAC significantly outperforms ICP

*Table 4.* Ablation study on the scale estimation and the registration algorithm.

| Method | SSIM↑ | PSNR↑ | LPIPS↓ | CLIP↑ |
|---|---|---|---|---|
| ICP | 0.88 | 16.08 | 0.28 | 0.66 |
| w/o scale factor estimation | 0.82 | 16.17 | 0.29 | 0.83 |
| Full Model | **0.89** | **17.00** | **0.20** | **0.97** |

in both success rate and registration accuracy.

## 6 Conclusion

In this paper, we introduced HIVE-3D, a novel framework for generating high-fidelity 3D scenes via a hierarchical voxel super-resolution approach. Taking an image and an initial coarse scene from TRELLIS as input, our method first constructs a scene tree to represent the hierarchical structure of the scene. This hierarchy then guides a coarse-to-fine synthesis process that populates the scene with detailed instances. Crucially, our approach preserves the coherent spatial layout of the initial scene while significantly enhancing it with high-resolution, high-fidelity components. Extensive experiments have demonstrated that HIVE-3D outperforms existing methods in terms of 3D scene generation quality.

Nevertheless, our method still has several limitations. First, our approach is based on TRELLIS, the resolution of each voxel generation is constrained and it struggles with large-scale scenes. Second, constructing the initial 2D hierarchical semantics relies on detection and segmentation models, which may introduce errors and lead to suboptimal results.

Future work will investigate extending our framework to multi-view and video-based 3D scene generation. In particular, we aim to explore joint cross-attention mechanisms that enable Scene Tree nodes to aggregate visual information from multiple viewpoints conditioned on camera poses. We also plan to incorporate multi-view consistency constraints into the intermediate stages of the diffusion process to further improve spatial coherence and temporal stability. We believe these extensions will enhance occlusion reasoning and facilitate the generation of dynamic 3D scenes in more complex real-world environments.

## Acknowledgements

This work was partially supported by National Key R&D Program of China (No. 2025YFG0101300), NSFC (No. 52572504), Key R&D Program of Zhejiang Province (No. 2025C01064), and NSFC Corporate Joint Key Program (No. U22B2034).

## Impact Statement

This work presents a method for high-quality 3D scene generation from a single image. The proposed approach could help accelerate 3D content creation for applications such as digital entertainment, virtual reality, and interactive design.

As with other generative AI technologies, our method could potentially be misused to create misleading or synthetic digital content. However, we believe the primary impact of this work is to support creative applications and improve the accessibility of 3D content creation tools.

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

# A  Additional Implementation Details

## A.1  Details on the attention computation

To perform instance-level segmentation directly on the initial scene voxels, we draw inspiration from Fuse3D(Jin et al., 2025) and first feed the complete set of tokens extracted from the image into $\mathcal{G}_{\mathrm{L}}$ of TRELLIS (Xiang et al., 2025) to derive the latent features for initial voxels $\{\boldsymbol{p}_i\}_{i=1}^{L}$. We obtain the final attention map by averaging the voxel-to-token attention maps from a specific subset of heads, namely head 0, 4, and 12. This map, softmax-normalized along the image token axis, yields an alignment score for each voxel, quantifying its correspondence with the visual content of the image. We then aggregate the scores of image tokens corresponding to the masked region $\{\mathcal{M}_k\}_{k=1}^{K}$. Finally, we select the set of voxels whose aggregated scores exceed a predefined threshold.

## A.2  Details on scale factor estimation.

We observe that the regenerated object $\mathcal{O}_k$ often exhibits significant discrepancies in scale and pose when compared to the original object. For scale estimation, a straightforward approach is to align the axial lengths based on the objects' bounding boxes. However, this method proves to be infeasible. The substantial pose difference between the two objects leads to considerable variations in the shapes of their respective bounding boxes, rendering any alignment based on them fundamentally unreliable. At the same time, we note that even when the pose of the object undergoes drastic changes, its overall geometry remains largely consistent, with refinements primarily in the surface details. Therefore, we base our scale estimation on this geometric invariance. Our method involves uniformly sampling a point cloud from the generated object's surface, computing the centroid of this point cloud, and then calculating the mean distance from all points to this centroid. This distance serves as a robust proxy for the overall size of the object. Finally, we define the scale factor $s$ as $d/d'$, where $d'$ is the size of the regenerated object $\mathcal{O}_k$ and $d$ is the size of the original object. We then rescale the object $\mathcal{O}_k$ by $s$ along each axis.

## A.3  Details on RANSAC-based Point Cloud Registration

In Sec. 4.3, we employ RANSAC-based global point cloud registration to robustly align the refined instance with its initial instance under geometric noise, partial shape discrepancies, and outliers. Specifically, we use the `registration_ransac_based_on_feature_matching` function in Open3D (Zhou et al., 2018), together with `TransformationEstimationPointToPoint` without scaling, to estimate a rigid transformation in SE(3).

Given the point clouds of the refined instance and the initial instance, we first downsample both point clouds using a voxel size of **0.0156**, corresponding to $1/64$ of the normalized unit space. This downsampling step reduces computational cost while preserving the overall geometric structure of each instance. Surface normals are then estimated using a hybrid search radius of **0.0312**, i.e., $2\times$ the voxel size. We further compute Fast Point Feature Histograms (FPFH) descriptors with a search radius of **0.0780**, i.e., $5\times$ the voxel size. For both normal estimation and FPFH computation, the neighborhood search is limited to a maximum of **200** nearest neighbors.

Based on the extracted FPFH descriptors, tentative feature correspondences are established between the two downsampled point clouds. We enable mutual filtering to retain correspondences that are mutually consistent in feature space, which helps suppress ambiguous or one-sided matches. During registration, the maximum correspondence distance is set to **0.0234**, corresponding to $1.5\times$ the voxel size. This threshold defines the maximum Euclidean distance allowed between a transformed source point and its target correspondence for the correspondence to be considered geometrically valid.

To further reject outlier correspondences, we use two correspondence checkers. The first is `CorrespondenceCheckerBasedOnEdgeLength` with a similarity threshold of **0.9**, which enforces approximate preservation of pairwise distances between sampled correspondences and rejects matches that would induce severe geometric distortion. The second is `CorrespondenceCheckerBasedOnDistance`, which uses the same distance threshold of **0.0234** to ensure that the transformed source points remain sufficiently close to their target correspondences.

The RANSAC procedure is configured with a minimal sample size of **3**, a maximum of **100,000** iterations, and a confidence level of **0.999**. Here, the minimal sample size of 3 refers to the number of point correspondences sampled to estimate a 6-DoF rigid transformation under the point-to-point alignment objective. It should not be interpreted as the number of points required to define a geometric plane. Given three non-collinear point correspondences, a rigid transformation can be estimated by solving for the rotation and translation, e.g., via the Kabsch algorithm using singular value decomposition (SVD). RANSAC

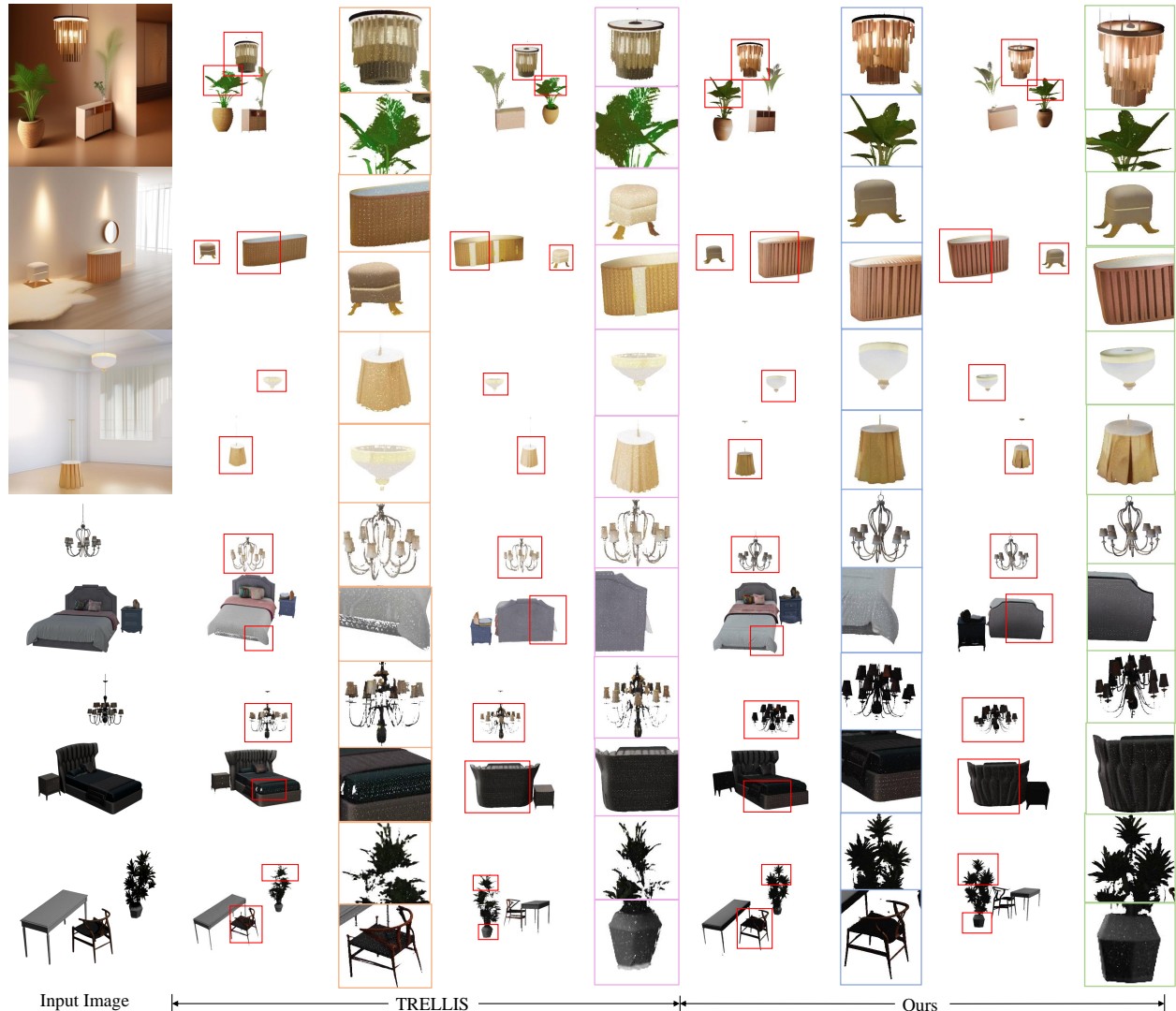

*Figure 8.* More results on synthetic data.

repeatedly samples such minimal correspondence sets, generates candidate rigid transformation hypotheses, and selects the transformation with the largest set of geometrically consistent inlier correspondences.

The output of this process is a rigid transformation matrix

$$\mathbf{T} = \begin{bmatrix} \mathbf{R} & \boldsymbol{t} \\ \mathbf{0}^\top & 1 \end{bmatrix} \in \mathrm{SE}(3),$$

where $\mathbf{R} \in \mathrm{SO}(3)$ is the estimated rotation matrix and $\boldsymbol{t} \in \mathbb{R}^3$ is the translation vector. The estimated transformation aligns the refined instance to the coordinate frame of the initial instance while suppressing unreliable correspondences caused by geometric noise and outliers. These parameter settings provide stable and reproducible global registration across diverse object instances.

### A.4 Training Details.

In practice, we employ the AdamW optimizer (Loshchilov & Hutter, 2017) for optimizing all networks and parameters. We set the initial learning rate to $1 \times 10^{-4}$ and the weight decay to $1 \times 10^{-2}$. We trained our model for approximately 3 days on 4 NVIDIA RTX 4090 GPUs, with a total batch size of 24. The total number of training iterations is set to 100,000. We

*Table 5.* Runtime Analysis.

| Nodes | Seg. + SR (s) | Generation (s) |
|---|---|---|
| 1 (depth = 1) | 0.00 | 59.3 |
| 2 (depth = 2) | 8.45 | 86.6 |
| 3 (depth = 2) | 13.13 | 92.6 |
| 4 (depth = 2) | 19.35 | 98.1 |

*Table 6.* Quantitative Comparisons on 3D-FRONT and real data.

| Method | SSIM↑ | PSNR↑ | LPIPS↓ | CLIP↑ |
|---|---|---|---|---|
| Gen3DSR | 0.76 | 10.96 | 0.34 | 0.91 |
| MIDI | 0.80 | 12.85 | **0.25** | 0.92 |
| SceneGen | 0.79 | 12.25 | 0.31 | 0.96 |
| TRELLIS | **0.80** | 13.31 | 0.31 | 0.95 |
| Ours | 0.79 | **13.39** | 0.33 | **0.97** |

apply Classifier-Free Guidance (CFG) (Ho & Salimans, 2022) training with a condition dropout rate of $10\%$. To ensure training stability, we maintain an Exponential Moving Average (EMA) (Polyak & Juditsky, 1992) of model parameters with a decay rate of 0.9999.

To construct our dataset, we curated a subset of 10,000 3D models from Objaverse-XL (Deitke et al., 2023), excluding those with excessive mesh complexity. Subsequently, the obtained meshes are voxelized into a grid with a resolution of $64^3$. We then employ the sparse structure VAE encoder from TRELLIS to encode these voxel grids into a feature volume of size $16 \times 16 \times 16 \times 8$. Finally, following the design of IP-Adapter (Ye et al., 2023), we reshape this volume into a sequence of 16 tokens, each with a dimensionality of 2048, serving as the coarse voxel condition for the training network.

### A.5 Runtime Analysis

We evaluate the computational efficiency of our proposed pipeline by analyzing the inference latency across various scene complexities. Specifically, we categorize the runtime performance according to the number of nodes in the scene tree (a single root at depth 1, or 2–4 nodes when expanding to depth 2). All benchmarks were conducted on a single NVIDIA GeForce RTX 4090 GPU (The total end-to-end execution time, including model loading and environment initialization), and the detailed results are summarized in Table 5.

## B Quantitative Results on Geometry

### B.1 Comparison Results.

We quantitatively evaluate the geometric quality of the scenes generated by our method and the baseline approaches on a synthetic dataset. The evaluation is conducted using SSIM, PSNR, LPIPS (Zhang et al., 2018), CLIP (Radford et al., 2021). As shown in Table 6, HIVE-3D demonstrates consistently strong performance across all metrics.

### B.2 Ablation Study on Geometry.

We conducted ablation studies on the synthetic dataset to evaluate geometric quality. We employed CD, F-Score, and IoU as evaluation metrics. The results are presented in Table 7. The results demonstrate that every core component contributes significantly to the overall performance of the pipeline.

## C More Experiments

### C.1 More Comparison Results

We evaluated our method on synthetic data. We render the final images from two viewpoints: one aligned with the input perspective and the other from the opposite viewpoint. For comparison, we also include the results rendered by TRELLIS

*Table 7.* Ablation study on Geometry.

| Method | CD↓ | F-Score↑ | IoU↑ |
|---|---|---|---|
| w/o voxel SR model | 0.0098 | 60.09 | 0.6188 |
| ICP | 0.147 | 23.92 | 0.2932 |
| w/o scale factor estimation | 0.193 | 35.44 | 0.2352 |
| Full Model | **0.0035** | **84.34** | **0.7449** |

from the corresponding viewpoints. Additional results are presented in Figure 8. Please zoom in to view the fine details.

## C.2  Ablation Study on Hierarchical Scene Generation

To further justify our design choices, we provide an in-depth analysis of the multi-level recursive structure. Unlike a conventional two-layer architecture—which typically transitions directly from an initial scene to fine-grained components—our approach employs a recursive refinement strategy to address the inherent resolution constraints of the TRELLIS framework.

Since TRELLIS generates scenes within a fixed $64^3$ canonical voxel space, directly partitioning the scene into the finest components would lead to an extreme jump in spatial resolution. For instance, a small component originally occupying only $4^3$ voxels in the initial scene would be abruptly re-represented in a $64^3$ space. This sharp transition introduces significant geometric ambiguity and compromises the precision of spatial registration, as evidenced by the misaligned car in Figure 9. In contrast, our recursive design facilitates a more progressive transition (e.g., $4^3 \rightarrow 16^3 \rightarrow 64^3$), providing a stable geometric bridge that ensures more reliable alignment and superior fidelity in the final scene assembly.

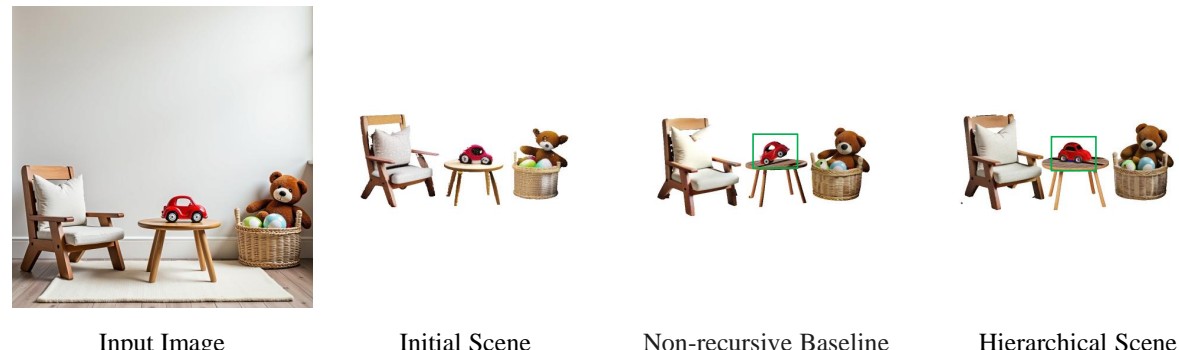

| Input Image | Initial Scene | Non-recursive Baseline | Hierarchical Scene |

*Figure 9.* Comparison between the direct two-layer partitioning and our multi-level recursive refinement.

## C.3  Qualitative Comparison with PartPacker

We conduct qualitative comparisons between our method and PartPacker (Tang et al., 2026). It should be noted that PartPacker primarily focuses on geometric synthesis and generates only untextured meshes. As illustrated in Figure 10, PartPacker exhibits several structural deficiencies across various scenarios. Specifically, it fails to reconstruct the correct number of table legs and struggles with plant stems in the first row. Furthermore, it suffers from object omissions and significant shape distortions in the third row, where the generated chair deviates notably from the input reference. In contrast, our HIVE-3D consistently produces scenes with both geometric precision and high-fidelity textures, underscoring the clear advantages of our hierarchical refinement strategy in complex scene synthesis.

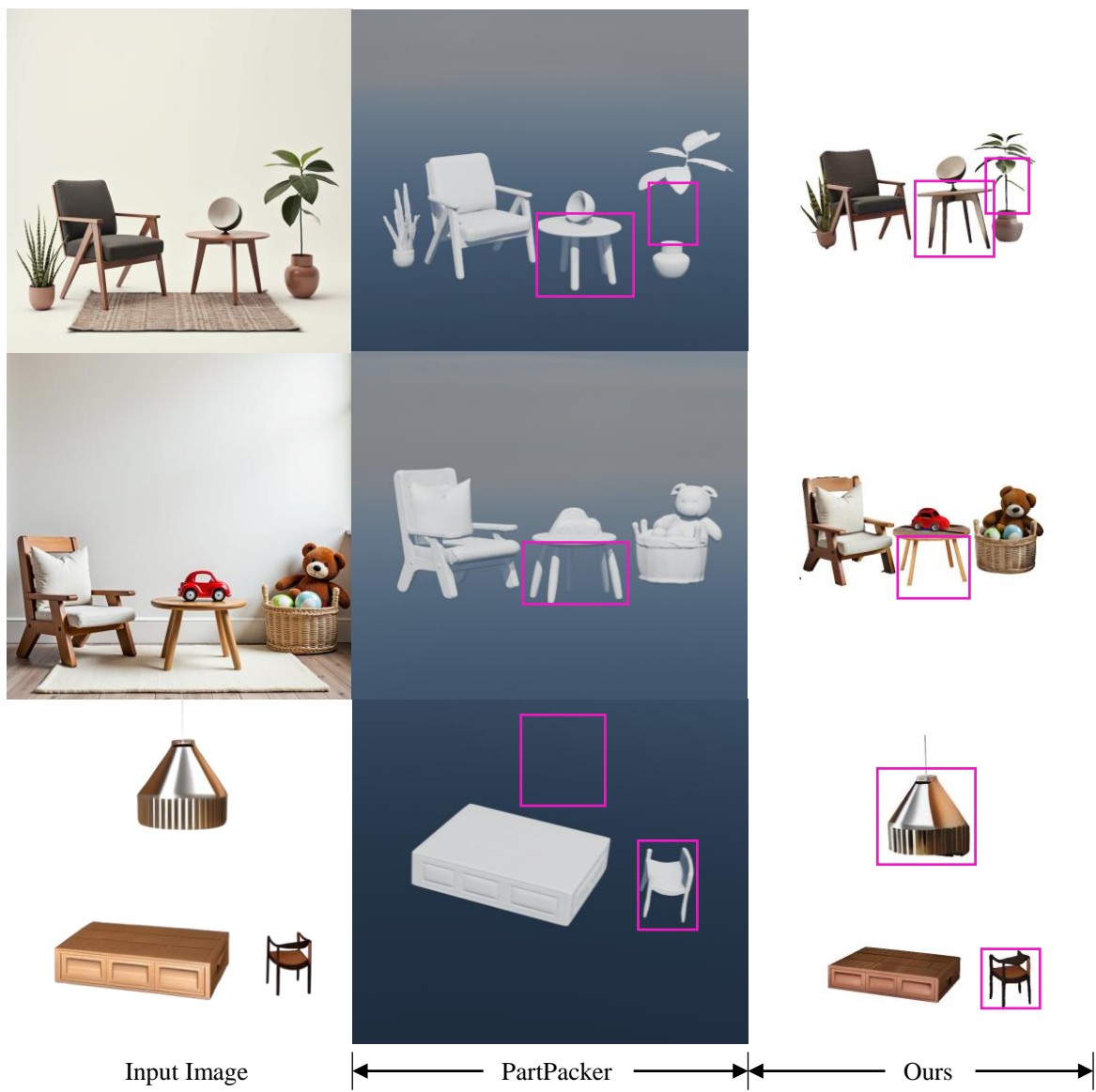

*Figure 10.* Qualitative comparison with PartPacker.

## C.4 Qualitative Comparison with SAM3D

We conduct qualitative comparisons between SAM3D (Chen et al., 2025a) and our method in terms of 3D scene generation quality. Since SAM3D performs holistic scene generation, its results are constrained by the overall scene resolution. As shown in the first row of Figure 11, the generated small stool exhibits noticeable geometric inconsistencies with the reference image. In the second row, the generated red toy car is incorrectly fused with the small table, leading to severe structural artifacts. In contrast, HIVE-3D produces scenes with substantially higher local geometric fidelity and more coherent object structures.

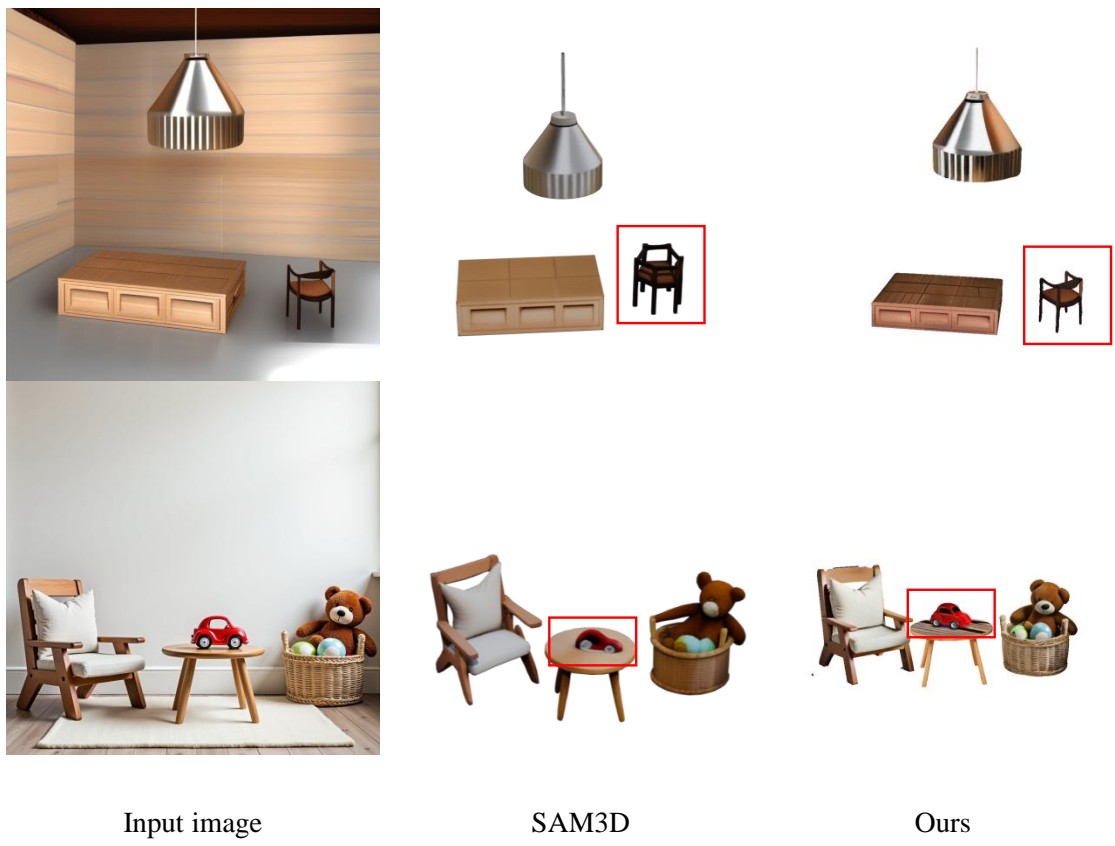

Input image                    SAM3D                    Ours

*Figure 11.* Qualitative comparison with SAM3D.

## C.5    Comparison with VIAFormer and ULTRA3D

Although VIAFormer (Fang et al., 2026) and ULTRA3D (Chen et al., 2025b) are also related to structured 3D generation, our method differs substantially from these approaches in both formulation and design objectives.

VIAFormer primarily focuses on denoising and refining existing occupancy representations under a fixed and stable spatial resolution. In contrast, our proposed Voxel-SR module performs generative voxel super-resolution, progressively enhancing sparse scene nodes into higher-resolution and higher-fidelity 3D structures within a hierarchical coarse-to-fine generation framework. Therefore, our method is designed not only for refinement, but also for hierarchical spatial upsampling and detail synthesis.

Compared with ULTRA3D, which relies on specialized part-aware sensing modules for semantic decomposition and alignment, our framework adopts a lightweight 2D-to-3D semantic lifting strategy that directly reuses intrinsic cross-attention maps from the diffusion model for semantic grounding. This design avoids introducing additional part sensing networks or expensive supervision, resulting in a more integrated and computationally efficient pipeline.

Overall, these distinctions enable HIVE-3D to achieve high-fidelity scene generation while maintaining strong semantic consistency and efficient hierarchical generation capabilities.

## D    Failure Analysis

Since our pipeline depends on the scene generated by the TRELLIS, its performance is inherently bounded by the quality of the initial output. In cases where the base model fails to generate certain objects or produces incomplete geometry, downstream components of the pipeline may struggle to function correctly. Specifically, this leads to failures in the registration stage, preventing refined components from being accurately placed in their global coordinates, as illustrated in

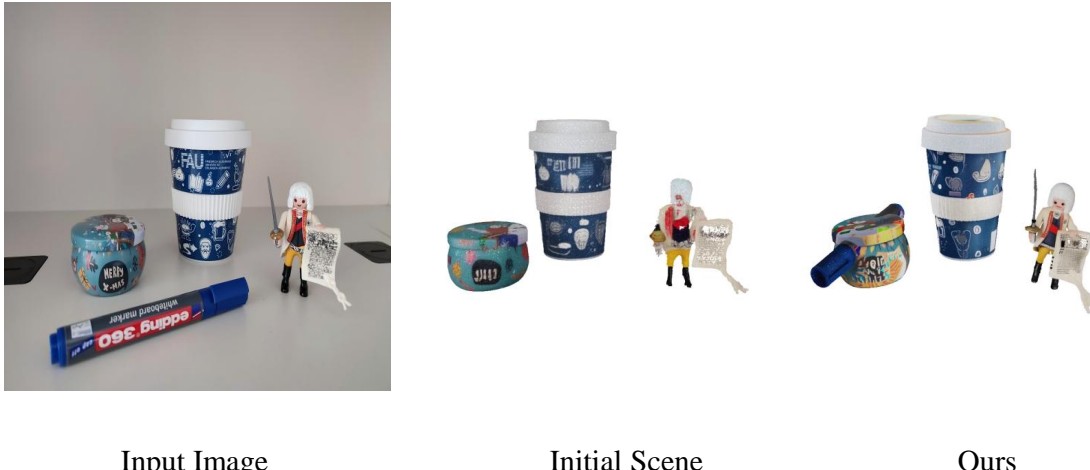

Input Image             Initial Scene             Ours

*Figure 12.* Representative failure case of our pipeline.

Figure 12. We aim to address this limitation by enhancing the robustness of the initial scene parsing in future work.

