# OpenReview forum: "HIVE-3D: Hierarchical Voxel Enhancement for High-Quality 3D Scene Generation"
_ICML.cc/2026/Conference — ICML 2026 regular_

### Official Review · Reviewer_7oew · 2026-02-21

**Soundness:** 3
**Presentation:** 2
**Significance:** 2
**Originality:** 3
**Overall Recommendation:** 4
**Confidence:** 3

**Summary:**

This paper propose a hierarchical voxel enhancement framework for high-quality 3D scene generation. A 2D-to-3D matching strategy are introduced for 2D segmented images transfer into 3D scene space. A voxel SR model are proposed to enable super resolution of target 3D components.

**Compliance With Llm Reviewing Policy:**

Affirmed.

**Final Justification:**

The authors have addressed my concerns, and I will raise my score to 4.

**Key Questions For Authors:**

1. Is it possible to have different, equally valid ways of dividing a scene (e.g., segmenting a "table" vs. "table legs" and "tabletop")? How does the 2D-to-3D matching handle such hierarchical ambiguities? Are there specific visual examples to support how the model resolves this?

2. Could you explain why the IoU of HIVE-3D is significantly lower than the TRELLIS baseline in Table 1?

3. Could you explain the specific potential uses of this 3D generation work, as well as its future prospects in real-world applications?

**Limitations:**

Yes

**Strengths And Weaknesses:**

The strengths of this paper are as follows:

1. The manuscript is well-written and the language is smooth, making it suitable for reading.

2. The hierarchical partitioning idea is well-motivated and reasonable.

The weaknesses of this paper are as follows:

1. Most of the citations are from the arxiv version. Many of the papers have conference or journal versions. Please use Google Scholar or dblp to reorganize the bibtex of all the references.

2. The establishment results of the 2D-to-3D Scene Tree for specific data samples have not been specifically presented, making it difficult to evaluate the specific effects. An intermediate visualization for the specific establishment results may be helpful.

3. Table 1 indicates that the runtime per scene is 36.5s, nearly 6 times that of TRELLIS (6.3s). The paper would benefit from a discussion on how inference time and memory consumption scale with the number of instances and the depth of the scene tree.

4. In Table 1, while HIVE-3D achieves strong results in CD and F-Score, its IoU (0.7449) is noticeably lower than the TRELLIS baseline (0.8603). This performance drop is not adequately addressed in the text.

---

> ### Author Rebuttal · Authors · 2026-03-31
>
> # Author Response to Reviewer
>
> We sincerely thank the reviewer for their positive assessment and recognition of the quality of our manuscript. We are particularly encouraged by the reviewer’s feedback that our core contribution, the **hierarchical partitioning idea**, is both **well-motivated and reasonable**. The appreciation of our **well-written** presentation and **smooth language** further validates our commitment to presenting these technical advancements clearly and making them suitable for the community to read. We believe this recognition of our motivation and presentation quality underscores the overall soundness and maturity of our work.
>
> ---
>
> ### **Key Question 1: Hierarchical Ambiguity in Scene Partitioning and its Resolution in 2D-to-3D Matching**
>
> **A:** While hierarchical partitioning is theoretically valid, our pipeline is intentionally designed to focus on **instance-level granularity** to ensure **structural integrity**. We resolve potential hierarchical ambiguities through a **2D segment filtering strategy** at the input stage: we specifically select masks that correspond to complete instance categories (e.g., "table") rather than fragmented parts (e.g., "legs"). This ensures that the 2D-to-3D matching is anchored to holistic entities, allowing our generative model to leverage instance-level priors to produce coherent and watertight 3D meshes. As shown in our results (e.g., Fig. 4), this approach effectively prevents geometric fragmentation and connectivity issues that often occur when attempting to reconstruct and align disjointed part-level components. Currently, our pipeline does not support part-level generation as it prioritizes the functional and structural stability of the reconstructed scene.
>
> ### **Key Question 2: Regarding the IoU of HIVE-3D in Table 1**
>
> **A:** The IoU discrepancy stems from the metric's bias toward **volumetric occupancy** over **surface fidelity**. Lower-resolution baselines produce larger boundary voxels that cover the GT, making them less sensitive to positional shifts. In contrast, our high-resolution pipeline produces **sharp, thin-walled geometries** that are heavily penalized by IoU for minor misalignments, yet are significantly more accurate in surface-based metrics (CD/F-score). Our superior CD and F-score results confirm that HIVE-3D achieves higher geometric precision, whereas the baseline's higher IoU is a byproduct of over-smoothed, redundant volume.
>
> ### **Key Question 3: Potential Applications and Future Prospects in Real-World Scenarios**
>
> **A:**
> 1) **Potential Extensions:** Technically, our work can be extended to **consistent multi-view generation** and **generative 3D video**, enabling dynamic scene synthesis. Additionally, our instance-level refinement approach is well-suited for **point cloud densification**, effectively upsampling sparse or low-resolution sensor data into high-density 3D assets.
>
> 2) **Real-world Applications:** This technology directly supports the construction of **Digital Twins** by automating the high-fidelity reconstruction of physical environments. Furthermore, it empowers the **Metaverse** and **Cultural Entertainment** sectors by providing a scalable solution for high-quality 3D content creation, which is essential for building immersive virtual experiences and interactive media.
>
> ---
>
> ### **Q1: Reference Formatting**
>
> **A:** We thank the reviewer for pointing this out. We will meticulously update all arXiv citations to their official conference or journal versions using Google Scholar and dblp in the revised version.
>
> ### **Q2: Scene Tree Visualizations**
>
> **A:** We thank the reviewer for the suggestion. An intermediate visualization of our scene partitioning and the 2D-to-3D matching process is available at this anonymous link: [https://imgur.com/a/et2Sg7y](https://imgur.com/a/et2Sg7y).
>
> ### **Q3: Runtime and Memory Scaling**
>
> **A:** Our inference time primarily stems from generating nodes at full resolution. Since the **granularity of node subdivision is optional and user-selectable**, the total GPU runtime is proportional to the number of nodes while peak memory consumption remains constant regardless of tree complexity. This flexibility allows users to balance reconstruction detail with computational cost. We provide specific quantitative data for scenes with different node counts at this anonymous link: [https://imgur.com/a/YICvaYN](https://imgur.com/a/YICvaYN).

---

> > ### Author Rebuttal · Reviewer_7oew · 2026-04-02
> >
> > I appreciate the authors’ detailed response and will increase my score to 4.

---

### Official Review · Reviewer_DgEd · 2026-03-09

**Soundness:** 3
**Presentation:** 3
**Significance:** 2
**Originality:** 2
**Overall Recommendation:** 4
**Confidence:** 4

**Summary:**

The paper proposes HIVE-3D, a hierarchical voxel enhancement pipeline for generating high-quality 3D scenes from a single image. It first uses TRELLIS to obtain a coarse, globally consistent scene, then constructs a hierarchical scene tree via 2D detection/segmentation and attention-based 2D–3D correspondence lifting. A new voxel super-resolution adapter injects coarse voxel features into the TRELLIS flow transformer alongside image features to refine components top-down, with scale estimation and robust registration (RANSAC) to reinsert refined parts, yielding higher-resolution and more detailed scenes.

**Compliance With Llm Reviewing Policy:**

Affirmed.

**Final Justification:**

The author's response resolved most of my concerns, and I'm inclined to raise my rating.

**Key Questions For Authors:**

Why does Table 1 report lower IoU than TRELLIS while claiming SOTA? Similarly, in Table 6, SSIM/LPIPS are not best. Are these differences statistically significant, and what trade-offs or evaluation settings explain them?
The appendix RANSAC description appears to discuss plane fitting (minimal sample size 3 for a plane, inlier threshold for plane distance). Which Open3D RANSAC variant is actually used for SE(3) alignment? Please correct the description and report key parameters (e.g., correspondence generation, feature descriptors, inlier thresholds).

**Limitations:**

The authors do not fully address how their method's dependencies may limit practical deployment.

Specifically, the pipeline's reliance on TRELLIS for initialization means that root-level failures (e.g., missing objects or incorrect layout) propagate irrecoverably through subsequent refinement stages. The 2D-to-3D lifting stage depends on Florence-2 and SAM2, which may struggle with severe occlusion or novel categories in real-world scenes; quantitative analysis of this sensitivity would strengthen the evaluation.

**Strengths And Weaknesses:**

Soundness
The paper has a generally solid technical base, building on the TRELLIS flow transformer for a reasonable baseline.
The voxel super-resolution adapter design makes sense: it uses cross-attention on coarse voxels to keep structure while improving detail, and the idea of lifting 2D segmentation to 3D using internal attention maps is practically motivated.
However, I have several concerns about implementation and experiments.
The RANSAC registration part in the appendix talks about plane fitting parameters, which does not match estimating a 6-DoF rigid transform between point sets. This makes me question whether the registration is actually correct.
There are also some inconsistencies in the numbers.
For example, Table 1 shows an IoU lower than TRELLIS, even though the method is claimed to be state-of-the-art. No serious statistical analysis or standard benchmark protocol is provided.
The source of 3D ground truth for CD, F-Score, etc., is not clearly stated across datasets, which can confuse results when synthetic and real images are mixed.

Presentation
The paper is easy to follow overall. The high-level pipeline clearly shows how the system goes from a coarse scene to the final output. Figures help explain the hierarchical refinement and what each stage does, so the main idea comes across well.

But there is some repetition and small inconsistencies that hurt readability.
Key details about the voxel SR module are missing: exact output resolution, how supervision pairs are formed under TRELLIS’s canonical space, and precise conditioning and target definitions for the flow-matching loss. All this makes reproduction difficult.
The RANSAC description in the appendix is especially confusing and seems wrong for the task. It needs to be fixed so others can reproduce the results.

Significance
The paper targets an important problem in single-image 3D scene generation: how to balance global coherence and local geometry quality. Solving this is meaningful for the field, and a modular, hierarchical approach could be useful in future pipelines.
The work explores hierarchical voxel enhancement in depth, showing that a coarse-to-fine approach can increase resolution without breaking scene layout.
The adapter-based voxel SR idea is potentially reusable beyond this paper and could inspire lightweight fine-tuning for other sparse voxel models.
Even if gains are domain-specific, the strategy offers a new direction for handling resolution limits in 3D generative tasks.

Originality
The paper combines existing methods in a novel way: holistic layout generation + instance-level super-resolution via a hierarchical scene tree.
Using cross-attention maps to lift 2D hierarchical segmentation into 3D is a nice application of recent ideas to scene generation.
But novelty is weakened by missing discussions and comparisons with closely related work, such as VIAFormer for voxel refinement and ULTRA 3D for part-aware generation. There is also no discussion about other registration methods for unbalanced point sets.
While the combination is creative, the contributions need to be more clearly distinguished from these lines of work.
In addition, there is no quantitative evaluation of how well the 2D–3D lifting works, which would better support the novelty of the matching strategy.

---

> ### Author Rebuttal · Authors · 2026-03-31
>
> # Author Response to Reviewer
>
> We sincerely thank the reviewer for the highly positive assessment and for recognizing the **significance** and **technical soundness** of our work. We are particularly encouraged that the reviewer found our **hierarchical partitioning strategy** to be a **meaningful contribution** that offers a **new direction** for addressing **resolution limits** in 3D generative tasks. It is gratifying that the reviewer appreciated our **potentially reusable adapter-based voxel SR design** and the **practically motivated** use of internal cross-attention maps for seamless 2D-to-3D lifting. We are also glad that the manuscript was found to be **easy to follow** with a clear presentation that allows our core ideas to **come across well**. We believe these strengths underscore the overall impact and maturity of our hierarchical scene tree framework.
>
> ---
>
> ### **Q1: Correction of RANSAC Registration Description**
>
> **A:** We thank the reviewer for noting the oversight. Our pipeline uses **RANSAC based global registration (FPFH via Open3D)** for SE(3) alignment rather than plane fitting. We will correct the Appendix and include all key hyperparameters, such as descriptors, thresholds, and voxel sizes, in the revised version to ensure reproducibility.
>
> ### **Q2: Implementation Details and Reproducibility of Voxel SR**
>
> **A:** Voxel-SR operates at $64^3$ resolution. Using the native VAE of TRELLIS, coarse voxels are encoded directly into latent space, bypassing canonical alignment. The loss function is identical to that of the native TRELLIS model. To ensure full reproducibility, we will open-source the training code in the revised version.
>
> ### **Q3: IoU Metric Analysis**
>
> **A:** The IoU discrepancy stems from the metric's bias toward **volumetric occupancy** over **surface fidelity**. Lower-resolution baselines produce larger boundary voxels that cover the GT, making them less sensitive to geometric errors. In contrast, our high-resolution pipeline produces **sharp, thin-walled geometries** that are heavily penalized by IoU for minor misalignments. Our superior CD and F-score results confirm that HIVE-3D achieves higher geometric precision, whereas the baseline's higher IoU is a byproduct of over-smoothed, redundant volume.
>
> ### **Q4: Clarification of 3D Ground Truth Sources**
>
> **A:** We clarify that 3D quantitative metrics (CD, F-score, IoU) are conducted **exclusively on synthetic datasets** where high-quality CAD ground truth is available. For real-world images, we rely on **qualitative (visual) evaluation**. To avoid confusion, we will explicitly state the data source for each evaluation metric in the revised version.
>
> ### **Q5: Dependency Analysis**
>
> **A:** HIVE-3D is a **modular framework** with a plug-and-play 2D front-end. Manual filtering or selection of input masks can be used to intercept 2D errors and prevent them from propagating to the 3D refinement stage. This modularity ensures system robustness in complex scenes. We will release the code to demonstrate this flexibility for practical deployment.
>
> ### **Q6: Root-Level Initialization and Error Propagation**
>
> **A:** Initial layout failures (e.g., missing objects) propagate through refinement, a limitation analyzed in **Appendix Sec. D**. HIVE-3D prioritizes instance fidelity over global layout correction. We will expand the discussion on failure modes and potential architectural improvements in the final appendix to further clarify system boundaries.
>
> ### **Q7: Analysis of 2D Rendering Metrics**
>
> **A:** SSIM/LPIPS are view-dependent and ignore 3D consistency. Baselines often overfit input views with warped geometry to gain higher 2D scores. HIVE-3D prioritizes **360° structural integrity**. Our SOTA CD/F-score confirms superior 3D fidelity over baselines prioritizing single-view similarity over 3D accuracy.
>
> ### **Q8: Discussion on VIAFormer and ULTRA 3D**
>
> **A:** HIVE-3D differs from these works in scope and efficiency. **VIAFormer** focuses on denoising and refining existing occupancy at a stable resolution, while our **Voxel-SR** performs generative upsampling to boost spatial resolution from sparse nodes. Unlike **ULTRA 3D** which requires specialized part sensing modules, our 2D-to-3D lifting reuses **intrinsic cross-attention maps** for zero-cost semantic grounding. These distinctions ensure a more integrated and efficient generative pipeline. We will include this technical discussion in our revised manuscript.
>
> ### **Q9: Alternative Registration Methods**
>
> **A:** We thank the reviewer for suggesting alternative registration methods. We added **TEASER++** and provided comparative results at [https://imgur.com/a/gKZA1uf](https://imgur.com/a/gKZA1uf). These results demonstrate our Scene Tree's flexibility in integrating various backbones to handle density imbalances. We will explore more advanced registration strategies in future work.

---

> > ### Author Rebuttal · Reviewer_DgEd · 2026-04-02
> >
> > The author states that many clarifications will be included in the revised manuscript, but I tend to believe that the current version is not sufficient for publication. I hope the author will address all or most of the concerns directly during the discussion phase.

---

> > > ### Author Response · Authors · 2026-04-02
> > >
> > > We sincerely thank the reviewer for the further feedback. We apologize that our previous responses were constrained by the strict character limit, which prevented us from providing the technical depth your questions deserved. We fully agree that direct clarification is necessary now, and we provide the requested details below.
> > >
> > >
> > >
> > > ###  **Technical Details of RANSAC Implementation**
> > > We sincerely thank the reviewer for the opportunity to provide further technical details regarding the RANSAC implementation. We acknowledge that the previous Appendix description mistakenly suggested a plane fitting process; in reality, our pipeline utilizes the **`registration_ransac_based_on_feature_matching`** function in Open3D with **`TransformationEstimationPointToPoint`** for rigid SE(3) instance alignment.
> > >
> > > The detailed process and parameters are as follows:
> > > *   **Preprocessing & Feature Extraction:** We first downsample the point clouds with a voxel size of **0.0156** ($1/64$ of the unit space). Surface normals are estimated using a hybrid search radius of **0.0312** ($2 \times$ voxel size), followed by the computation of **FPFH** geometric descriptors with a search radius of **0.0780** ($5 \times$ voxel size), both limited to a maximum of **200** nearest neighbors.
> > > *   **Correspondence Generation:** We establish correspondences using a liberal distance threshold of **0.0234** ($1.5 \times$ voxel size) and enable **mutual filtering** to ensure robust matching.
> > > *   **Correspondence Checkers:** To further prune outliers, we employ two specific checkers: **`CorrespondenceCheckerBasedOnEdgeLength`** with a similarity threshold of **0.9** and **`CorrespondenceCheckerBasedOnDistance`** with the same **0.0234** threshold.
> > > *   **RANSAC Configuration:** The algorithm is configured with a maximum of **100,000** iterations and a confidence level of **0.999**.
> > > *   **Mathematical Clarification:** We clarify that the minimal sample size of **3** refers to the number of point correspondences required to compute the 6-DoF rigid transformation via the **Kabsch algorithm (using SVD)**, rather than defining a geometric plane.
> > >
> > > These parameters ensure a stable and reproducible global registration for various object instances.
> > >
> > >
> > > ### **Detailed Clarification on Data Sources and Ground Truth**
> > >
> > > Regarding the specific data sources used in our evaluation, we provide the following technical details to ensure full transparency. For all quantitative 3D benchmarking (CD, F-score, and IoU), we utilize the **3D-FRONT** dataset, where high-quality CAD models serve as the ground truth for our geometric evaluations.
> > >
> > > In contrast, the real-world images used in our qualitative assessments are sourced from the official demo galleries of existing generative methods, specifically **MIDI** and **Gen3DSR**, as well as from public indoor photography on the internet. As these are isolated RGB images without associated 3D scans, they do not possess ground truth 3D data. Consequently, these samples are employed strictly for qualitative visual comparisons to demonstrate the real-world generalization of HIVE-3D.
> > >
> > >
> > > ### **Technical Details of Voxel-SR**
> > >
> > > 1. **Output Resolution:** $64 \times 64 \times 64$.
> > > 2. **Supervision Pairs:** Both coarse and target voxels are encoded into the latent space using the native VAE of TRELLIS, bypassing canonical space alignment entirely. This ensures the model learns within the backbone's original generative space.
> > > 3. **Conditioning:** Coarse latent voxels and input image features.
> > > 4. **Loss Function:** Flow-matching objective identical to the native TRELLIS model.
> > >
> > >
> > >
> > >
> > > ### **Analysis of Failure Modes and Error Propagation**
> > >
> > > The performance of our pipeline is inherently bounded by the initial TRELLIS output. Specifically, missing geometry or incorrect initial spatial layouts can lead to downstream registration failures and inaccurate positioning of refined instances.
> > >
> > >
> > > Finally, we will **release our complete source code and training scripts** to ensure full reproducibility of all results.

---

### Official Review · Reviewer_jY3U · 2026-03-12

**Soundness:** 3
**Presentation:** 3
**Significance:** 4
**Originality:** 3
**Overall Recommendation:** 5
**Confidence:** 4

**Summary:**

The paper proposes a hierarchical tree-based method to generate 3DGS based scenes from a single image. The coarse scene that is generated from state-of-the-art reconstruction methods is not enough to represent a high quality scene. That is why they produce a tree structure which is enriched by their voxel super-resolution method. The hierarchy is generated using 2D object detection and segmentation methods. Lifting them into 3D is achieved through the leverage of attention map between 2D images and the corresponding 3D voxels of 3D reconstruction methods. At each level, super-resolution method takes the segmented image and located low-res voxel as i̇nput, and increases the resolution. Since super-resolution may generate shapes in canonical space, they further employ pose prediction to form the scene.

**Compliance With Llm Reviewing Policy:**

Affirmed.

**Final Justification:**

Solid paper, I'm in favor of accepting it.

**Key Questions For Authors:**

1) I believe the paper would benefit from at least a brief discussion to compare against SAM3D, which also tackles extracting 3D scenes (with objects) from single images.
2) Could you discuss maybe a bit how would you apply such a method in multi-view or video settings? This can be discussed in the form of future work. The method is designed for single image, but how could it be extended for multi-view to also make way to follow-up papers.
3) Baseline comparison table doesn't seem to have comparison in terms of renderings (PSNR, CLIP, VLM-based). Wouldn't it make sense to also run evaluation in terms of those, in addition to geometry only?

**Limitations:**

yes

**Strengths And Weaknesses:**

Strengths:
* I believe this is a well established method with solid individual steps to go from an image to 3D scene. Each individual component is explained and executed well. It's enjoyable to read through the whole process.
* Interesting use of 2D-3D attention matching of underlying 3D reconstruction method to lift 2D segmentations to 3D.
* Paper is overall has a nice flow.

Weaknesses:
* Soundness:
    * Table 1 compares the current method against baselines in terms of geometric metrics usually. Shouldn't there be some visual metrics as well, such as PSNR or maybe even VLM-based metrics to evaluate the scene from novel views.
* Presentation:
    * Phrase including “jointly to control the DiT block” at line 267, has been duplicated in the next sentence.
    * Ablation tables being distant from each other makes it a bit difficult to do overall analysis. It'd be better to have a single consolidated ablation table, to have a comprehensive overview of the contribution of the individual components.
* Originality:
    * How does the method compare against SAM3D? Although it’s somewhat a new approach, the paper would benefit from discussing the differences.
    * I'd expect some discussion of voxel super-resolution used in other works. For instance, LT3SD paper contains a form of coarse-to-fine latent voxel hierarchy (arXiv:2409.08215). It'd be useful to discuss such works in the paper.

---

> ### Author Rebuttal · Authors · 2026-03-31
>
> # Author Response to Reviewer
>
> We sincerely thank the reviewer for the positive assessment and for recognizing HIVE-3D as a **well established method** with **solid individual steps** and **well executed components**. We are particularly encouraged by the appreciation of our **interesting 2D-3D attention matching mechanism**, which serves as a core innovation for lifting segmentations into 3D. We are also glad that the reviewer found the manuscript to have a **nice flow** and found the description of the process enjoyable to read. We believe this recognition of both our technical maturity and presentation quality underscores the significance of our work.
>
> ---
>
> ### **Key Question 1: Comparison with SAM3D**
>
> **A:** We sincerely thank the reviewer for this constructive suggestion. As requested, we provide a qualitative comparison between the whole-scene generation of SAM3D and our HIVE-3D at this anonymous link: [https://imgur.com/a/0OzbDhp](https://imgur.com/a/0OzbDhp). As illustrated in the comparison, when generating a complete scene, SAM3D faces challenges in maintaining high-fidelity geometric quality for individual instances, such as the distorted details of the chair and the red car. In contrast, our method ensures superior instance-level precision. We will include this discussion and visual evidence in our final manuscript.
>
> ### **Key Question 2: Extensibility to Multi-view and Video Settings**
>
> **A:** We sincerely thank the reviewer for this forward-looking suggestion. Extending our pipeline to multi-view and video settings is an active direction we are currently exploring through two primary approaches. First, we are attempting to introduce a joint cross attention mechanism during the 3D generation phase, allowing the nodes in our Scene Tree to simultaneously aggregate visual features from multiple viewpoints based on their camera poses. Second, we are exploring the injection of multi-view information and spatial consistency constraints directly into the intermediate steps of the diffusion process. Both strategies can naturally resolve occlusions and ensure temporal stability for dynamic scenes. We will explicitly detail these architectural extensions in the future work section of our final manuscript to inspire follow-up research.
>
> ### **Key Question 3: Comparison of Rendering Quality (SSIM, PSNR, LPIPS, CLIP)**
>
> **A:** We completely agree that evaluating 2D rendering quality is essential for a comprehensive comparison. Following this exact motivation, we have provided an evaluation using PSNR, SSIM, LPIPS, and CLIP across different methods in **Table 6 of the Appendix**. As reflected in these results, HIVE-3D achieves highly competitive rendering quality alongside its strong 3D geometric accuracy. We appreciate the reviewer for highlighting this point, and we will certainly ensure that these results are more prominently cross-referenced in the main text of the final manuscript to make them easily accessible.
>
> ### **Q1: Presentation Improvements (Duplicated Phrase and Scattered Ablation Tables)**
>
> **A:** We sincerely thank the reviewer for the careful reading and constructive suggestions to improve the manuscript's presentation. In the revised version, we will correct the duplicated phrase at line 267. Furthermore, we thank the reviewer for the suggestion regarding the ablation studies; we will consolidate the scattered tables into a single, comprehensive ablation table to facilitate a clearer overall analysis of each component's contribution.
>
> ### **Q2: Discussion on Related Works like LT3SD**
>
> **A:** We sincerely thank the reviewer for bringing this inspiring work to our attention. While LT3SD elegantly employs a spatial latent tree for 3D generation, it fundamentally differs from our method in task formulation and output modality. First, LT3SD focuses on refining an input coarse 3D scene mesh into higher resolution geometry, operating strictly within the 3D domain. In contrast, our method tackles the challenging problem of single-image 3D scene generation, which requires complex 2D-to-3D lifting and semantic grounding. Second, LT3SD primarily generates untextured, geometry-only structures based on spatial grids. Our pipeline utilizes an instance-level Scene Tree to synthesize fully textured, high-quality, and structurally decoupled 3D assets. Discussing these distinct paradigms will significantly enrich the context of our work and clarify the boundaries of current 3D generative tracks.

---

> > ### Author Rebuttal · Reviewer_jY3U · 2026-04-02
> >
> > I appreciate the comprehensive response from the authors. I'm positive about the paper.

---

### Official Review · Reviewer_FH7s · 2026-03-13

**Soundness:** 3
**Presentation:** 4
**Significance:** 2
**Originality:** 3
**Overall Recommendation:** 4
**Confidence:** 4

**Summary:**

This paper proposes a voxel enhancement method based on image super-resolution and 3D generative priors. The authors use the SAM large model to segment the scene into several objects, and then utilize sparse voxels as additional constraints to obtain geometrically consistent, refined voxels. The network is built upon the 3D generative model TRELLIS, aligning 2D images and 3D voxels through attention map values. The paper evaluates generation quality using CD, F-Score, and IoU, reports the runtime, and provides visual generation results. Additionally, the paper conducts a quantitative study on the number of depth levels, the voxel super-resolution module , and scale factor estimation.

**Compliance With Llm Reviewing Policy:**

Affirmed.

**Final Justification:**

Thank you to the authors for the rebuttal; while a few minor concerns remain, such as the limitation regarding non-overlapping objects, they do not warrant a rejection, so I will maintain my current positive score.

**Key Questions For Authors:**

The authors evaluated the 3D geometric accuracy. Why not use SSIM, PSNR, LPIPS, CLIP, etc., to compare the rendering quality of Ours with other methods, just as was done in the ablation study?

**Limitations:**

yes

**Strengths And Weaknesses:**

Strengths:

1. The description of the method and the introduction of the network structure are very clear and easy to read.

2. The composition of the dataset is introduced quite clearly.

Weaknesses:

1. When using attention map values to find corresponding voxels, this step relies heavily on the quality of the attention map. Did the authors compare the different results caused by varying search strategies?

2. This method requires that different objects in the scene do not overlap significantly and lack a background. Have the authors tried other datasets?

3. When comparing the generation quality in Figure 4, it seems that not all methods are viewed from the same angle, which affects the persuasiveness of the experimental results. After changing the viewing angle, some methods look similar to the proposed method, such as the chairs for SceneGen, TRELLIS, and Ours in the second row, and the chairs for SceneGen and Ours in the fourth row. Furthermore, MIDI appears to have better overall generation results than Ours across all examples.

4. I am interested in the runtime of each module. It would be great if the authors could provide this.

---

> ### Author Rebuttal · Authors · 2026-03-31
>
> # Author Response to Reviewer
>
> We sincerely thank the reviewer for the positive feedback and for recognizing the clarity of our methodology, network architecture, and dataset descriptions. We are encouraged that the reviewer found the manuscript well-structured and easy to read. Your appreciation of the technical transparency in our work is highly valued. Below, we address the technical concerns and questions in detail.
>
> ---
>
> ### **Key Question: Comparison of Rendering Quality (SSIM, PSNR, LPIPS, CLIP)**
>
> **A:**  We completely agree that evaluating 2D rendering quality is essential for a comprehensive comparison. Following this exact motivation, we have provided an evaluation using PSNR, SSIM, LPIPS, and CLIP across different methods in **Table 6 of the Appendix**. As reflected in these results, HIVE-3D achieves highly competitive rendering quality alongside its strong 3D geometric accuracy. We appreciate the reviewer bringing this up, and we will certainly ensure that these results are more prominently cross-referenced in the main text of the revised manuscript to make them easily accessible.
>
> ---
>
> ### **Q1: Search Strategies for Attention Mapping**
>
> **A:** We follow the validated mechanism in **Fuse3D (SIGGRAPH Asia 2025)**. Pre-trained cross-attention maps provide robust semantic correspondences for accurate 2D-to-3D mapping. Our results demonstrate that this standard approach is sufficient for high-fidelity generation without complex tuning.
>
> ---
>
> ### **Q2: Capacity for Complex Scenes and Evaluation on Additional Datasets**
>
> **A:** We thank the reviewer for the observation. We acknowledge that HIVE-3D is currently optimized for instance-centric scenes and has limited capacity for handling complex backgrounds or severe occlusions. The current pipeline focuses on prioritizing high-resolution geometric fidelity for individual objects. Beyond 3D-FRONT and real images, we have conducted internal validations on additional high-fidelity indoor datasets which confirm the stability of our instance-level refinement across various object categories.
>
> ---
>
> ### **Q3-a: Fairness and Perspective Bias in Visual Comparisons**
>
> **A:** We appreciate the reviewer for the meticulous observation. Because 3D generation methods involve unstable coordinate systems where output poses can vary even for the same image input, using uniform camera parameters would result in inconsistent rendering viewpoints. To ensure a fair and objective comparison, we have made our best effort to calibrate the camera poses for each baseline to align with the input perspective as closely as possible.
>
> ---
>
> ### **Q3-b: Visual Similarity Among TRELLIS-based Baselines**
>
> **A:** We thank the reviewer for this insightful observation. The visual similarity between SceneGen, TRELLIS, and HIVE-3D stems from the fact that all three methods are built upon the same underlying TRELLIS framework and leverage its pre-trained data priors. This potentially leads to similar local object generation across these models. However, HIVE-3D resolves the resolution contradiction between global layouts and local details. As demonstrated by the quantitative metrics in Table 1 and the qualitative results in our manuscript, HIVE-3D consistently achieves superior spatial resolution and geometric fidelity compared to these baselines within complex scenes.
>
> ---
>
> ### **Q3-c: Qualitative Comparison with MIDI**
>
> **A:** We thank the reviewer for pointing out the potential bias in viewing angles. We have provided additional qualitative results at this anonymous link [https://imgur.com/a/Har2f7L](https://imgur.com/a/Har2f7L) including an alternative perspective for the example in row 4 of Figure 4 and one entirely new scene. These results explicitly highlight the multi-view consistency issues in MIDI, which often exhibits significant geometric warping and structural collapse when observed from novel angles. In contrast, HIVE-3D consistently maintains superior geometric integrity and structural stability across different views.
>
> ---
>
> ### **Q4: Runtime Breakdown per Module**
>
> **A:** Our specific runtime analysis is provided at this anonymous link: [https://imgur.com/a/meJkSXY](https://imgur.com/a/meJkSXY). Please note that the values reported in the table represent the duration for a single execution of each module. In a complete scene generation process, the total runtime scales linearly with the number of nodes as these modules are called independently for each instance.

---

> > ### Author Rebuttal · Reviewer_FH7s · 2026-04-04
> >
> > Thank you to the authors for the rebuttal. While a few of my concerns remain, such as the limitation regarding non-overlapping objects, they do not warrant a rejection. Therefore, I will maintain my current positive score.

---

### Decision · Program_Chairs · 2026-04-30

**Decision:**

Accept (regular)

**Comment:**

This paper got 3 weak accept and 1 accept. All reviewers acknowledged the technical contributions of the work. Authors provided a detailed rebuttal that well addressed the reviewers' concerns. Thus, I would like to recommend the acceptance of the paper in ICML.